# Current Uses of Mushrooms in Cancer Treatment and Their Anticancer Mechanisms

**DOI:** 10.3390/ijms231810502

**Published:** 2022-09-10

**Authors:** Hye-Jin Park

**Affiliations:** Department of Food Science and Biotechnology, College of BioNano Technology, Gachon University, 1342 Seongnam-daero, Sujeong-gu, Seongnam-si 13120, Korea; nimpi79@hanmail.net

**Keywords:** cancer, mushrooms, mushroom-derived compounds, signaling pathways, multidrug resistance (MDR)

## Abstract

Cancer is the leading cause of mortality worldwide. Various chemotherapeutic drugs have been extensively used for cancer treatment. However, current anticancer drugs cause severe side effects and induce resistance. Therefore, the development of novel and effective anticancer agents with minimal or no side effects is important. Notably, natural compounds have been highlighted as anticancer drugs. Among them, many researchers have focused on mushrooms that have biological activities, including antitumor activity. The aim of this review is to discuss the anticancer potential of different mushrooms and the underlying molecular mechanisms. We provide information regarding the current clinical status and possible modes of molecular actions of various mushrooms and mushroom-derived compounds. This review will help researchers and clinicians in designing evidence-based preclinical and clinical studies to test the anticancer potential of mushrooms and their active compounds in different types of cancers.

## 1. Introduction

Cancer is one of the leading causes of deaths worldwide, and accounted for nearly 10 million deaths in 2020 according to WHO [1]. Accurate cancer diagnosis is important for effective treatment because each type of cancer requires a specific regimen. Cancer treatments are diverse, including surgery, anticancer drug treatment, radiotherapy, and systemic therapy (chemotherapy and targeted biological therapies). However, the current anticancer drugs available in the market are not target-specific, leading to the development of drug resistance, and even causing several side effects in clinical chemotherapy [2]. Therefore, it is important to develop novel and effective anticancer agents with low toxicity. In this regard, natural compounds have been highlighted as anticancer drugs. Mushrooms have been used in traditional medicines in East Asia due to their immunomodulatory, anticancer, and anti-inflammatory activities [2]. Among 14,000 different species of mushrooms, approximately 700 species have been reported to exhibit medicinal properties [3]. Recently, many studies have revealed the biological activities and the mechanisms of actions of mushroom compounds [2,3,4,5]. Some mushrooms and their active compounds possess anticancer properties. Polysaccharides isolated from *Phellinus linteus* (PLP) suppressed tumor growth and pulmonary metastasis through stimulating the immune response, not directly toxic to cancer cells [4]. Triterpenoids from *Ganoderma lucidum* showed anticancer properties [5]. β-D-glucans from *Ganoderma lucidum* exhibited anticancer effect by inhibiting cancer cells, protecting normal cells against free radicals, and reducing normal cell damage [2]. Their potential use as adjuncts in cancer therapy or as anticancer agents has emerged. Numerous clinical trials are in progress to assess the benefits of medicinal mushroom extracts in chemotherapy [2].

In this review, the clinical use and anticancer mechanisms of mushrooms are described. Our goal is to provide information pertaining to the potential therapeutic use of mushroom extracts and their active compounds against various cancers by elucidating the underlying targeted signaling pathways. Furthermore, these mushroom-derived compounds with anticancer activities can be exploited as novel anticancer agents. 

## 2. Uses of Mushrooms in Cancer Therapy

Many groups have reported that mushrooms possess anticancer activities and minimize undesirable side effects such as nausea, bone marrow suppression, anemia, and insomnia, and lower drug resistance after chemotherapy and radiation therapy [6] (Table 1 and Appendix A).

In the RCT conducted by Tsai et al., advanced adenocarcinoma patients treated with *Antrodia cinnamomea* alongside chemotherapy developed less severe gastrointestinal symptoms, such as abdominal pain and diarrhea, than those in the placebo group [7]. Twardowski et al. reported that *Agaricus blazei* Murill decreased prostate-specific antigen (PSA) levels and regulated recurrent prostate cancer by decreasing immunosuppressive factor [8]. Ahn et al. reported that patients with gynecological cancers receiving chemotherapy showed fewer side effects, such as loss of appetite, alopecia, and general weakness, when the therapy was accompanied by *Agaricus blazei* Murill compared to those in the placebo group [9]. Hetland et al. demonstrated that there was an increased number of plasmacytoid dendritic cells (pDC) and T regulatory cells (Tregs) in the blood; increased serum levels of IL-1Ra (receptor antagonist), IL-5, and IL-7; and an increased level of immunoglobulin genes, killer immunoglobulin receptor (KIR) genes, and human leukocyte (HLA) genes in the bone marrow in *Agaricus blazei* mushroom extract (AndoSanTM)-treated myeloma patients [10]. Loss of appetite decreased over time in patients that underwent six cycles of chemotherapy accompanied with *Agaricus sylvaticus*, while most patients in the placebo group suffered from loss of appetite and gastrointestinal symptoms, such as diarrhea, constipation, nausea, and vomiting [11]. In advanced lung cancer, 3–84% of patients receiving *Ganoderma lucidum* exhibited significantly improved cancer-related symptoms (e.g., fever, cough, weakness, sweating, and insomnia) compared to the placebo group (11–43%) [12]. In the randomized controlled trial (RCT) conducted by Zhao, breast cancer patients treated with *Ganoderma lucidum* showed less cancer-related fatigue than patients undergoing endocrine therapy [13]. In a phase I/II trial of breast cancer survivors, *Grifola frondosa* extract acted as an immunomodulator by increasing the production of IL-2, IL-10, TNF-α, and IFN-γ by subsets of T cells [14]. Hackman et al., reported that *Lentinula endodes* treatment alone was ineffective in treating prostate cancer patients [15]. The anticancer activity of a semisynthetic derivative of illudin S from Omphalotus illudens is due to the alkylation of DNA, RNA, and proteins. However, its use in the clinic is limited due to its strong retinal toxicity and narrow therapeutic index [16]. In a phase I trial conducted by Torkelson, *Trametes versicolor* enhanced the immune status in immunocompromised breast cancer patients [17]. In an RCT performed by Chay et al. for advanced hepatocellular carcinoma (HCC), patients treated with *Trametes versicolor* had a longer median overall survival (OS) and median progression-free survival compared to the placebo group [18]. The immunostimulatory effect and direct toxicity to cancer cells exhibited by *Trametes versicolor* polysaccharides implies that they can be applied as more than an adjuvant therapy [19]. 

## 3. Anticancer Compounds from Medicinal Mushrooms

The bioactive compounds found in mushrooms include polysaccharides, proteins, fats, phenolics, alkaloids, ergosterol, selenium, folate, enzymes, and organic acids. The anticancer components in mushrooms are antroquinonol, cordycepin, hispolon, lectin, krestin, polysaccharide, sulfated polysaccharide, lentinan, and Maitake D Fraction [20]. Many mushrooms are currently under clinical trials, and only a few are available for clinical use [2]. Polysaccharides are the most potent mushroom compounds with antitumor and immunomodulatory properties. Among polysaccharides, β-glucan consists of a backbone of glucose residues linked by β-(1→3)-glycosidic bonds, frequently with attached side-chain glucose residues joined by β-(1→6) linkages [21]. β-Glucan stimulates the immune system as a non-self molecule by inducing the production of cytokines that activate phagocytes and leukocytes [20]. Lentinan and lectins from *Lentinula edodes* have shown cytotoxic effects on breast cancer cells [22]. Lentinan from *Lentinula edodes* (also called Pyogo in Korea), schizophyllan (also called SPG, sonifilan, sizofiran, and sizofilan) from *Schizophyllum commune*, and PSK (also called krestin) from *Trametes versicolor* have been approved as prescription anticancer drugs in Japan [23]. Mushroom polysaccharides stimulate natural killer cells, T cells, B cells, and macrophages, leading to an increased immune response [24]. Cordycepin, also known as 3-deoxyadenosine, is a major anticancer compound in *Cordyceps* species. It exerts an apoptotic effect via dysregulated polyadenylation, and causes the termination of DNA or RNA elongation by binding to the site where nucleic acids are to be bound [25]. Hispolon, an active polyphenol compound, has been reported to exert potent antineoplastic properties and enhance the cytotoxicity of chemotherapeutic agents [26]. These findings suggest that some mushrooms may act synergistically in combination with commercial anticancer drugs as effective tools for treating drug-resistant cancers [6].

## 4. Anticancer Mechanisms of Medicinal Mushrooms

In Asian countries, medicinal use of mushrooms has been prevalent for a long time; however, in recent decades, their use for treating a number of diseases, including cancers, has increased in other parts of the world. The tremendous therapeutic potential of edible and medicinal mushrooms is attributed to the bioactive substances present in mushrooms. To increase the therapeutic success rates against cancer, it is important to understand the molecular mechanisms underlying cancer development and progression and the molecular targets of mushroom-derived bioactive compounds. In this section, we discuss how mushrooms help overcome multidrug resistance (MDR) and target signaling pathways, such as PI3K/AKT, Wnt-CTNNB1, and MAPK, during cancer treatment (Table 2). 

### 4.1. Overcoming Pgp-Mediated MDR Using Mushrooms

Drug resistance is a major obstacle in chemotherapy. Endogenous or acquired drug resistance represents the simultaneous development of resistance in tumor cells to drugs that are not mechanically or structurally related. This phenomenon is known as multidrug resistance (MDR). Resistance to anticancer drugs is one of the main factors of treatment failure, resulting in high morbidity. The overexpression of efflux pumps, which leads to multidrug resistance, is an important issue that needs to be resolved.

A major form of MDR is the overexpression of ATP-binding cassette (ABC) transporters and P-glycoprotein (Pgp), a 170 KDa transmembrane glycoprotein product encoded by the *MDR1* gene. The mechanism of drug resistance in Pgp-expressing tumor cells involves an increase in the extracellular transport of various chemotherapeutic agents, which diminishes cellular accumulation, and thus, decreases drug efficacy [31,44,45]. Anticancer agents with Pgp-mediated MDR include paclitaxel (TAX), doxorubicin (DOX), actinomycin D, vinblastine, and etoposide, whereas Pgp does not affect the cytotoxicity of certain other anticancer drugs, such as 5-fluorouracil, cisplatin, and carboplatin [46]. To confirm the drug resistance reversal activity of Basidiomycete mushroom extracts collected in Korea, the cytotoxic activity of paclitaxel (TAX), a well-known Pgp-related anticancer drug, on Pgp-positive and -negative human cancer cells in the presence or absence of the tested mushroom extract was compared to that in the presence of verapamil (VER), a well-known MDR reversal agent. *Cantharellus cibarius* (M02) and *Russula emetica* (M12) increased the cytotoxic activity of TAX by blocking Pgp-mediated drug efflux in Pgp-positive HCT15 and MES-SA/dX5 cancer cells, but not in Pgp-negative A549 and MES-SA cancer cells. *Cantharellus cibarius* and *Russula emetica* also increased the cytotoxicity of doxorubicin, another Pgp-associated anticancer drug, against MES-SA/DX5 cells [31]. *Ganoderma* species induced apoptosis in drug-sensitive (H69) and multidrug-resistant (VPA) human small-cell lung cancer (SCLC) cells that were resistant to etoposide and doxorubicin [47]. *Ganoderma lucidum* polysaccharide (PLP) inhibits the constitutive activation of NF-κB, decreasing the expression of PGP in cancer cells [48]. Zhankuic acids A–C isolated from *Taiwanofungus camphoratus* were found to exert inhibitory effects against Pgp, which reversed drug resistance against doxorubicin, vincritine, and paclitaxel in human cancer cells [43] (Figure 1). 

### 4.2. Overcoming Tumor Resistance to Inhibit Immune Checkpoint Interactions, the PD-1 Pathway, and CTLA-4/CD80, Using Mushrooms

In recent years, immune checkpoint blockade (ICB) therapy has caused a paradigm shift in cancer immunotherapy; it primarily inhibits various checkpoints that control host T cell activity by regulating the immune checkpoint interactions, PD-1/PD-L1 and CTLA-4/CD80. Programmed cell death-1 (PD-1) (CD279) is an inhibitory receptor, which is expressed in activated CD8+ T cells (as well as on B cells and natural killer cells) and leads to reduced innate and adaptive immune responses [49]. Particularly, PD-1 is highly expressed in tumor-specific T cells, which has prompted researchers to examine whether the inhibition of PD-1 suppresses cancer aggression by promoting an effective immune response [50,51]. The binding of PD-1 to the PD-L1 ligand allows cancer cells to evade the host immune response. In addition, PD-L1 induction protects cancer cells from T-cell-mediated destruction [52]. PD-1/PD-L1-checkpoint-blocking antibodies have been focused on as a powerful ICB therapy for cancer patients. However, patients with cancer quickly develop resistance to immunotherapy. β-Glucan from medicinal mushrooms, which acts as an immune adjuvant, has been found to stimulate innate and adaptive immune responses. It has been reported that administration of whole glucan particle (WGP) β-glucan along with PD-1/PD-L1-checkpoint-blocking antibodies leads to increased recruitment of immune-associated cells, improves the regulation of the balance between T cell activation and immune tolerance, and delays tumor progression [53]. This combination therapy was also found to improve progression-free survival in patients with advanced cancer who had previously discontinued anti-PD-1/PD-L1 therapy because of disease progression. These findings suggest that β-glucan can be used as an immune adjuvant to reverse anti-PD-1/PD-L1 resistance by regulating the immune system [54]. *Ganoderma lucidum* and its bioactive compounds reduced PD-1 protein level in cultured GM00130 and GM02248 human B-lymphocytes, thus, preventing and treating cancer [33] (Figure 2). 

The binding between CD80 on antigen-presenting cells (APCs) and CD28 on naive T cells results in T cell activation in the lymph nodes, which elevates the immune response and kills cancer cells. However, the interaction of CTLA-4 on naive T cells with CD80 on cancer cells produces an inhibitory signal for T cell activation, leading to the inhibition of T cell activation and suppression of the immune response [55]. Inonotus obliquus blocks CTLA-4/CD80 interaction and increases T cell activity so that cancer cells cannot escape the immune response [36] (Figure 2).

### 4.3. Targeting the PI3K/AKT Signaling Pathway in Cancer Using Mushrooms

The PI3K/AKT pathway is involved in the acquisition of chemotherapeutic drug resistance. The activation of phosphatidylinositol 3-kinase (PI3K) signaling leads to VEGF production, reduces tumor CD8+ T cell infiltration, and induces subsequent resistance to PD-1 blockade therapy. Currently, to avoid primary resistance to the PD-1/PD-L1 blockade, clinical treatment regimens that combine kinase inhibitor therapy with an immune checkpoint blockade are in use to enhance the response rates [52]. Activated phosphoinositide 3-kinase (PI3K) is a key signaling molecule that affects cell survival, proliferation, and differentiation by triggering the sequential activation of AKT and other downstream pathways [2,56]. When many components of this pathway are altered, it leads to the development of various cancers in humans [2].

Several research groups have demonstrated that mushroom-derived compounds can exert antitumor and antimetastatic effects by affecting various molecules in the PI3K/AKT pathway. For example, hispolon derived from *Phellinus linteus* inhibited the invasion and motility of a highly metastatic liver cancer cell line (SK-Hep1) by downregulating MMP2, MMP9, and uPa, and inhibiting the activation of the ERK1/2, PI3K/AKT, and FAK pathways [38]. Proteoglycan (P1) from *Phellinus linteus* showed antiproliferative activity in multiple human cancer cells by inducing a notable decrease in AKT, Reg IV, EGFR, and plasma PGE2 concentrations [57]. A polysaccharide–protein complex isolated from *Pleurotus pilmonarius* (PP) suppressed PI3K/AKT signaling in liver cancer cells [2]. Additionally, when PP was used in combination with cisplatin, the sensitization of liver cancer cells to cisplatin was improved. The phosphorylation of BAD at Ser 136 via AKT is required for cell viability. When this AKT node is suppressed in ovarian cancer cells, they become sensitive to cisplatin. Inhibition of PI3K/AKT signaling by PP made the cells more sensitive to cisplatin [2]. Ganoderic acid from *Ganoderma lucidum* suppressed human glioblastoma by inducing apoptosis and autophagy via the inactivation of the PI3K/AKT signaling pathway [58].

### 4.4. Targeting the Wnt/β-Catenin Pathway in Cancer Using Mushrooms

A high rate of abnormality in the Wnt signaling pathway has been observed in many cancers. APC, CTNNB1, AXIN1, FAM123B, and TCF7L2 are the key molecules in Wnt signaling pathway that may undergo somatic mutations related to common human cancers, including colon cancer [59]. The onset and progression of sporadic colon cancer (CRC) and familial adenomatous polyposis (FAP)-associated diseases are believed to be caused by mutations in the adenomatous polyposis coli (APC) gene [2]. Depending on the stage and type of cancer, the Wnt-CTNNB1 signaling pathway can either promote or inhibit tumor initiation, growth, metastasis, and drug resistance [2]. An inverse correlation has been reported between β-catenin/Wnt activation in cancer and the degree of CD8+ T cell infiltration in a mouse model of melanoma [60]. Increased β-catenin/Wnt activity also correlated with diminished CD103+ dendritic cell infiltration due to reduced levels of CCL4, a chemokine responsible for attracting them. PD-1 blockade therapy was ineffective in melanoma tumors with β-catenin/Wnt activation, whereas this treatment worked well in tumors without β-catenin/Wnt mutations [52,61]. Wnt inhibition increased tumor T cell infiltration and inhibited tumor proliferation and migration by enhancing PD-1 antibody treatment and upregulating the expression of PD-L1 in mice with glioblastoma (GBM) [62].

Several groups have reported the anti-oncogenic activities of different mushroom-derived compounds via Wnt–CTNNB1 signaling. For instance, 4-acetylantroquinonol and antroquinonol from *Antrodia camphorata* was discovered to inhibit colon cancer by suppressing the Wnt/β-catenin pathway [28,29]. *Antrodia camphorata* grown on germinated brown rice suppressed human colon cancer cell proliferation by upregulating G0/G1 phase arrest and apoptosis and reducing β-catenin signaling [40]. Researchers have shown that *Phellinus linteus* can inhibit tumor growth, invasion, and angiogenesis by downregulating genes (cyclin D1 and TCF/LEF) of the Wnt signaling pathway in SW480 human colon cancer cells as well as in vivo. *Phellinus linteus* grown on germinated brown rice attenuated the levels of NF-κB, β-catenin, and mitogen-activated protein kinase (MAPK) proteins [41]. Ergosterol peroxide and 4-acetylantroquinonol from *Inonotus obliquus* inhibited nuclear β-catenin in colon cancer cells [2]. When the level of β-catenin activity was reduced, the expression of β-catenin target genes (c-myc, cyclin D1, and VEGF) was also decreased, thus exerting anticancer effects on meningioma cells [63]. Therefore, these compounds may be potential candidates for pharmaceutical treatment of human meningiomas.

### 4.5. Targeting the MAPK Pathway in Cancer Using Mushrooms

Other mutations associated with T cell exclusion and subsequent resistance to PD-1/PD-L1 blockade usually occur within the mitogen-activated protein kinase (MAPK) signaling cascade. Constitutive oncogenic signaling activated through this pathway leads to the production of immunosuppressive cytokines, viz., vascular endothelial growth factor (VEGF) and interleukin 8 (IL-8), which inhibit T cell recruitment to the cancer tissue as well as their activity [52]. Mutations within the MAPK cascade are common in melanomas, and inhibition of this cascade is known to improve CD8+ T cell infiltration within cancers and sensitize them to PD-1 blockade therapy [52]. This result strongly suggests that a combination therapy involving multikinase inhibition with PD-1 blockade can be used in cancers with such mutations.

Platinum-based anticancer drugs have been shown to upregulate PD-L1 expression through the MAPK pathway in gastric cancer cells. A β-glucan from *Lentinula edodes*, viz., lentinan, suppressed cisplatin- or oxaliplatin-induced PD-L1 expression, suggesting that a combination of lentinan and platinum-based chemotherapy can recover the chemosensitivity of cells [64]. In addition, the MAPK pathway plays a pivotal role in oncogenesis, as several oncoproteins upstream of MAPK cascade, including ErbB-2, Scr RTKs, Ras, and Raf, are mutated into activated forms of enzymes [65]. The triterpene-enriched fraction, WEES-G6, from *Ganoderma lucidum* inhibited Huh-7 human hepatoma cell growth [66,67]. Yang et al. showed that *Antrodia camphorata* markedly inhibited the MAPK signaling pathway, thereby suppressing the invasion/migration of highly metastatic MDA-MB-231 cells [26,68]. Furthermore, *G. lucidum* triterpene extract (GLT) suppressed the phosphorylation of p38 MAPK, leading to antophagy in colon cancer cells [69]. Co-treatment using the extract of *Phellinus linteus* grown on germinated brown rice (PBR) and cetuximab reduced MAPK signaling by decreasing *KRAS* expression. PBR enhanced the sensitivity of *KRAS*-mutated colon cancer cells to cetuximab [40].

### 4.6. Targeting the NF-κB Pathway in Cancer Using Mushrooms

The nuclear transcription factor κB (NF-κB) is one of the factors responsible for cellular chemoresistance, and controls a myriad of gene expressions, including antigen receptors on immune cells, adhesion molecules, proinflammatory cytokines, and chemoattractants for inflammatory cells [70]. NF-κB is associated with neoplastic development, including insensitivity to growth inhibitory signals, avoidance of apoptosis, metastasis and sustained angiogenesis, and chronic inflammation [71].

Numerous studies highlight the antitumor effect of mushrooms through targeting the NF-κB signaling pathway. Antroquinonol (AQ) and 4-acetylantroquinonol B (4-AAQB) from *Antrodia Camphorata* exhibited inhibitory effects on NF-κB signaling in MCF-7 breast cancer cells [72]. Kadomatsu et al. found that treatment with cordycepin, a major compound of *Cordyceps militaris*, became sensitive to TNF-α-mediated apoptosis, which suppresses pro-survival NF-κB. Ho’s group reported that *Ganoderma* suppressed metastasis in highly invasive breast and prostate cancer cells by blocking constitutively active AP-1 and NF-κB signaling [73]. Treatment with sulfated polysaccharide obtained from *Grifola frondosa* (S-GFB) resulted in apoptosis of HepG2 cells through the induction of S phase arrest, inhibiting notch1 expression, degradation of IκB-α, translocation of NF-κB from the cytoplasm to the nucleus, and the activation of caspase 3 and 8 [35]. *Phellinus linteus* was shown to produce caffeic acid phenethyl ester (CAPE), which specifically inhibits NF-κB binding to DNA [74,75]. 

## 5. Regulating Immune Function in Cancer Using Medicinal Mushrooms

Despite the increasing success of existing personalized cancer treatments, recurrence and metastasis are common, depending on the type and stage of the disease [11]. Many studies have reported the beneficial effects of medicinal mushrooms, which particularly enhance the quality of life and reduce the side effects of conventional chemotherapy. In addition, their positive effects on anticancer activity and immune regulation have been reported. Several mechanisms have been suggested for the antiproliferative and immunomodulatory effects of medicinal mushrooms [2,20,76,77]. Mushroom polysaccharides stimulate dormant natural killer cells, T cells, B cells, and macrophage-dependent immune responses [24].

Mushroom-derived compounds activate immune cells to induce either cell-mediated or direct cytotoxicity in cancer cells by binding to pathogen recognition receptors. Compounds, such as lentinan, increase the proliferation of cytotoxic T lymphocytes and macrophages and induce nonspecific immune responses [78]. *Pleurotus tuber* and *Pleurotus rhinoceros* extracts were shown to promote the activation of lymphocytes and NK cells and increase macrophage proliferation, T helper cell number, and CD4/CD8 ratio and population, conferring anticancer effects [9,11,20,79]. Natural killer cells act as the key cells in innate immunity by attacking major histocompatibility class I-negative target cells that can evade immune surveillance of cytotoxic T cells. The activity of natural killer cells in patients with gynecological cancer undergoing chemotherapy was significantly enhanced when co-treated with *Agaricus blazei* Murill for 3 to 6 weeks as compared to that in the placebo group [9]. Leukopenia results in cachexia and metabolic changes in cancer patients and increases the risk of infection [80]. In patients with multiple myeloma treated with *Agaricus blazei* Murill, the immune status was much better in terms of maintaining the population of white blood cells and immunoglobins and led to fewer infections [81]. The mushroom’s main component, β-glucan, exerts hematopoietic effects and increases bone marrow regeneration in vitro [82]. β-Glucan also significantly increased the population of DCs (CD11c+/CD8+) and macrophages (CD11b+/F4-80+) and decreased the population of regulatory T cells and myeloid-derived suppressor cells (MDSC)s, resulting in an enhanced immune response [54]. *Ganoderma lucidum* supplementation resulted in a more stable disease state in the lung cancer study population than in the control group [83]. In addition, there was a significant increase in CD3 percentage, natural killer cell activity, and lymphocyte mitogenic reactivity against concanavalin A in lung cancer patients [11]. *Cordyceps militaris* fermented with *Pediococcus pentosaceus* (GRC-ON89A) treatment was reported to aid the recovery of immune activity in high-dose cyclophosphamide (a chemotherapeutic drug)-treated mice by increasing the phagocytic activity of mouse peritoneal macrophages and stimulating NO production in macrophages. GRC-ON89A reduced the toxicity of anticancer agents through the recovery of the immune system [84]. 

Thus, mushroom-derived compounds induce innate and adaptive immunity by enhancing immune surveillance against cancer by affecting monocytes, macrophages, NK cells, and B cells, and by activating immune organs [85,86], which leads to cancer cell apoptosis, cell cycle arrest, and prevention of angiogenesis and metastasis [20]. Consumption of mushroom compounds also boosts the secretion of antitumor cytokines by CTLs and activation of immune organs, thereby eliminating cancer cells and strengthening the weakened immune system [87].

## 6. Prebiotic Properties of Medicinal Mushrooms in Cancer

Several groups have reported that medicinal mushrooms can act as prebiotics, and thus enhance the growth of beneficial microbiota. Prebiotics can affect the human intestinal microbial population and suppress various diseases such as diabetes, obesity, and cancer [77]. The important sources of prebiotics in mushrooms are polysaccharides, such as chitin, hemicellulose, β- and α-glucans, mannans, xylans, and galactans, which can suppress the proliferation of pathogens by increasing the growth of probiotics in the gut [6,88]. A poor intestinal microbiota composition can lead to the development of cancer [88]. It is possible that prebiotic effects medicinal mushrooms could enhance quality of life (QOL) during and after cancer therapy.

## 7. Conclusions

This review demonstrates the potential use of mushrooms and their anticancer mechanisms in cancer treatment. Mushroom-derived bioactive compounds activate and/or regulate the immune system by affecting the maturation, differentiation, and proliferation of immune cells, thereby inhibiting cancer cell metastasis and growth. It is very important to understand the underlying mechanisms of action of the anticancer compounds derived from mushrooms to suppress cancer and improve the quality of life of cancer patients. Mushrooms show anticancer potential by regulating a single molecule of a specific signaling pathway, or by having multiple targets in the same or different signaling pathway(s), including the PI3K/Akt, Wnt/β-catenin, and MAPK pathways. In addition, several studies have highlighted the effect of mushroom-derived components as single and adjuvant therapeutic agents in reversing MDR by targeting Pgp, PD-1/PD-L1, and CTLA-4/CD80 interactions. In addition, the prebiotic effects of medicinal mushrooms could enhance quality of life (QOL) during and after cancer therapy by recovering the intestinal microbiota.

However, only a few clinical studies of a small number of mushrooms demonstrate the positive effects of medicinal mushrooms, including reductions in the adverse effects of conventional therapies, as well as antitumor activity and immunomodulation. Therefore, more clinical research on mushrooms with anticancer potential needs to be conducted, especially by employing high-quality methodology, larger sample sizes, standard mushroom preparations, and long-term follow-ups. In addition, future studies should investigate the preventive aspects of medicinal mushrooms in reducing the rate of cancer occurrence by being a part of a healthy diet and lifestyle. High-quality clinical studies are needed to identify the potential of medicinal mushrooms in cancer treatment.

## Figures and Tables

**Figure 1 ijms-23-10502-f001:**
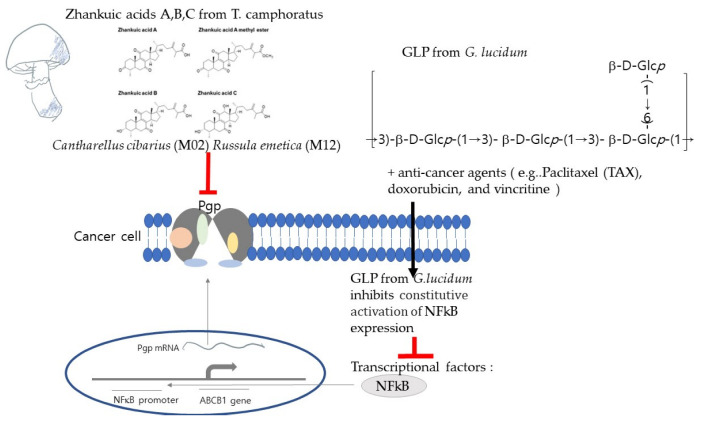
An illustration of the anticancer mechanisms of mushrooms by overcoming Pgp-mediated MDR. Structure of zhankuic acids A, B, and C from *T. camphoratus* adapted from Zhankuic Acids A, B and C from Taiwanofungus Camphoratus Act as Cytotoxicity Enhancers by Regulating P-Glycoprotein in Multi-Drug Resistant Cancer Cells. Biomolecules, 2019. **9**(12).

**Figure 2 ijms-23-10502-f002:**
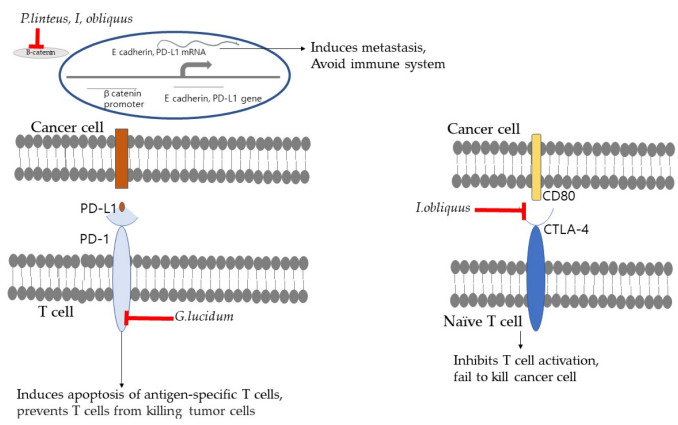
The illustration of the anticancer mechanisms of mushrooms by inhibiting the PD-1 pathway and CTLA-4/CD80 pathway.

**Table 1 ijms-23-10502-t001:** Ongoing clinical trials for use of mushrooms in cancer treatment.

Mushroom	Cancer	Phase	Study Status	Active Compound/s	Identifier	Investigator
*Agaricus bisporus*	Breast cancer, cancer survivors	Phase 1	Completed	Polysaccharides, lectin	NCT00709020	Shiuan Chen
*Agaricus bisporus*	Prostate cancer	Phase 1b	completed	Polysaccharides, lectin	NCT00779168	W. Twardowski
*Agaricus blazei* Murill (AndoSan)	Multiple myeloma	Phase 2	Completed	Agaricus polysaccharides	NCT00970021	Jon-Magnus Tangen
*Lentinula edodes*	Prostate cancer	Not mentioned	Completed	Genistein Combined Polysaccharide (GCP)	NCT00269555	Robert Hackman
*Lentinula edode*	Hepatocellular Carcinoma and Hepatitis B and C Infection	Not mentioned	Completed	Arabinoxylan extracted from rice bran treated enzymatically with extract from *Lentinula edode*	NCT01018381	Mai Hong Bang
*Grifola frondosa*	Lung Neoplasms and Breast Carcinoma	Phase 1	Completed	Polysaccharides	NCT02603016	Shunchang Jiao
*Omphalotus illudens*	Thyroid Cancer	Phase 2	Completed	A semisynthetic derivative of illudin S	NCT00124527	Eisai Inc.
*Omphalotus illudens*	Recurrent or Metastatic Gastric Cancer	Phase 2	Completed	A semisynthetic derivative of illudin S	NCT00062257	Winnie Yeo
*Omphalotus illudens*	Recurrent or Persistent Epithelial Ovarian Cancer	Phase 2	Completed	A semisynthetic derivative of illudin S	NCT00019552	Gisele A. Sarosy
*Trametes versicolor*	Breast cancer	Phase 1	Completed	Krestin, PSK, PSP	NCT00680667	Carolyn Torkelson

**Table 2 ijms-23-10502-t002:** Anticancer mechanisms of mushrooms and their bioactive substances.

Mushroom Species/Reference	Bioactive Substance	Experimental Study		Target/Mechanism
*Antrodia camphorata* [27]	4-Acetylantroquinonol B	Colorectal cancer	DLD-1, HCT-116, SW-480, RKO, HT-29	Lgr5/Wnt/β-catenin, JAK–STAT↓
*Antrodia camphorata* [21]	Polysaccharide (ACE)	Hepatocellular carcinoma	HepG2 cell line	Apoptosis
*Antrodia camphorata* [9]	Antroquinonol	Pancreatic carcinoma	PANC-1 and AsPC-1 cells	AKT at p-Ser 473↓ mTOR at p-Ser 2448↓
*Antrodia camphorata* [28]	Antroquinonol	Colon cancer	HCT15, HCT-116 and LoVo cells	PI3K/AKT/β-catenin signaling↓
*Antrodia camphorata grown on germinated brown rice (CBR)* [29]	Adenosine	Melanoma		MITF and TRP-1↑, p53↑
*Antrodia camphorata grown on germinated brown rice (CBR)* [30]		Colon cancer		β-catenin pathway↓
*Cantharellus cibarius* [31]				Drug resistance in Pgp-expressing tumor cells↓
*Cordyceps militaris* [32]	Cordycepin		NRK-52E cell line	NF-κB↓
*Ganoderma lucidum* [33]				PD-1 protein↓
*Ganoderma lucidum* [34]	Polysaccharide	Liver cancer	HepG2, Bel-7404	p27kip↑, cyclinD1/CDK4↓, cyclin E/CDK2↓, AKT at p-Thr 308 and p-Ser 473↓, pPTEN↑, Bcl-2 activation, apoptosis, caspase 3 and 9↑
*Grifola frondose* [35]	Sulfated polysaccharide	Liver cancer	HepG2	Apoptosis, S phase arrest, NOTCH1↓, IκB-α degradation, FLIP↓, Caspase 3 and 8↑
*Inonotus obliquus* [36]	Lanosterol, terpenoid			CTLA-4/CD80 interaction↓ Activation of T cells↑
*Inonotus obliquus* [37]	Ergosterol peroxide	Colorectal cancer	HCT116, HT-29, SW620, DLD-1 CRC cell lines	β-catenin pathway↓
*Phellinus linteus* [38]	Hispolon	Human hepatoma cells	SK-Hep1 cells	MMP2↓, MMP9↓, uPA↓, p-ERK1/2, p-PI3K/AKT↓, p-FAK↓
*Phellinus linteus* [39]	Protein-bound polysaccharide	Colon cancer	SW480 cells	Wnt/β-catenin Pathway↓, Cyclin D1↓, TCF/LEF↓
*Phellinus linteus* [34]	Polysaccharide	Liver cancer	HepG2, Bel-7404	p27kip↑, cyclinD1/CDK4↓, cyclin E/CDK2↓, AKT at p-Thr 308 and p-Ser 473↓, pPTEN↑, Bcl-2 activation, apoptosis, caspase 3 and 9↑
*Phellinus linteus grown on germinated brown rice (PBR)* [40]	Not determined	KRAS-mutated colon cancer		MAPK pathway↓
*Phellinus linteus grown on germinated brown rice (PBR)* [41]	γ-Aminobutyric Acid and β-glucan	colon cancer metastasized to the lung		NF-κB, β-catenin, MAPK pathway↓, MMP2 and 9 activities↓
*Phellinus linteus grown on Panax ginseng (PGP)* [42]	Rd, Rg1, Re, Rb2, and Rg3	melanoma		Caspase 8 and 9, p53 and p21 ↑
*Phellinus linteus grown on Panax ginseng (PGP)* [30]	Not determined	Colon cancer		Caspase 8 and 9↑
*Russula emetic* [31]				Drug resistance in Pgp-expressing tumor cells↓
*Taiwanofungus camphoratus* [43]	Zhankuic acids A–C			Drug resistance in Pgp-expressing tumor cells↓

Note: ↑, upregulation; ↓, downregulation; p, phosphorylation; NG, not given.

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
