# Peer review of "Current Uses of Mushrooms in Cancer Treatment and Their Anticancer Mechanisms"

_ijms, 2022, doi:10.3390/ijms231810502_

Round 1

Reviewer 1 Report

After critical evaluation, it can be said that the entitled paper "Current usages of mushrooms in cancer treatment and their anti-cancer mechanisms" is interesting as cancer resistance is growing up. Natural products are comparatively safer than synthetic drugs. 

This paper is well written. However, references need to cite appropriately in the first few sentences of the introduction (lines 23-29). The author needs to increase the contents of the introduction to highlight the specific problems and with justification how Mushroom can solve the current problems. 

Remove the heading results, as this is not a systematic review. 

Add illustrated molecular mechanisms of mushroom products for anti-cancer actions, and how do mushrooms overcome the Pgp-mediated multidrug resistance? Illustrate too. 

The author can show a common illustrated molecular mechanism showing the different signalling pathways. 

The conclusion should be concise and there should be no need to use citations unless it is necessary. 

Author Response

For Reviewer 1

We greatly appreciate the efforts of the Editorial Board and reviewers for giving us valuable comments on our manuscript for publication in IJMS. Here, we have enclosed a revised version of the manuscript in response to the reviewer’s comments. Included below is a point-by-point description of our responses to the reviewer’s comments. The revision made in the manuscript was written in blue. We cordially hope that the reviewers find this revised manuscript acceptable for publication in the special issue ‘Role of NF-κB in Carcinogenesis and Its Therapeutic Regulation’ of IJMS

Sincerely,

Hye-Jin Park

Reviewer 2 Report

The manuscript entitled “Current usages of mushrooms in cancer treatment and their anti-cancer mechanisms” has been reviewed. The selected topic is noteworthy, which could have a higher scientific interest; however, the current manuscript has some considerable deficiencies:

·        The manuscript does not achieve the main goal of a review article, i.e. to present a clear and thorough review, an overall picture of the selected topic.  It seems to be rather a compilation of cited articles without a well-defined structure presented in a logical way

·        In spite of the fact that the selected topic is quite manageable with only few clinical trials available dealing with the anticancer potential of mushrooms the manuscript does not mention several important clinical trials, for e.g.:

Preventing Recurrence of Hepatocellular Carcinoma After Curative Hepatectomy With Active Hexose-correlated Compound Derived From Lentinula edodes Mycelia. By: Kamiyama et al., Integrative cancer therapies (2022)

Reduction of Adverse Effects by a Mushroom Product, Active Hexose Correlated Compound (AHCC) in Patients With Advanced Cancer During Chemotherapy-The Significance of the Levels of HHV-6 DNA in Saliva as a Surrogate Biomarker During Chemotherapy. By: Ito, Toshinori et al., Nutrition and Cancer (2014), 66(3), 377-382

Efficacy and safety of orally administered Lentinula edodes mycelia extract for patients undergoing cancer chemotherapy: a pilot study. By: Yamaguchi et al. The American journal of Chinese medicine (2011), 39(3), 451-9

Dietary administration of mushroom mycelium extracts in patients with early stage prostate cancers managed expectantly: a phase II study. By: Sumiyoshi et al. Japanese journal of clinical oncology (2010), 40(10), 967-72

A phase I/II trial of a polysaccharide extract from Grifola frondosa (Maitake mushroom) in breast cancer patients: immunological effects. By: Deng et al. Journal of Cancer Research and Clinical Oncology (2009), 135(9), 1215-1221.

·        if we are talking about the potential role of mushrooms and mushroom derived compounds in cancer therapy then we should mention irofulven, a semisynthetic derivative of illudine S from Omphalotus illudens. Irofulven was studied in numerous clinical trials in the early 2000s, but it was ultimately abandoned because it worked on only a small percentage of patients and had several side effects. Now irofulven is again in the focus and hopefully the vast experience of the previous clinical trials will not be lost.

·        It is important to emphasize that most of the mushrooms with beta glucan content used in the treatment of different types of cancer do not have a direct anticancer activity but rather possess an immunomodulatory property making them as good candidates in the adjuvant therapy of cancer to alleviate the negative side effects caused by chemo- and radiotherapy

·        I miss the chemical formula of the fungal metabolites responsible for the observed anticancer activity of the discussed mushrooms

·        I would suggest a better fitting title, which could reflect the potential of mushrooms in anticancer treatment

·        The overall scientific value of the manuscript is not high enough, the novelty is debatable

Some minor points:

  • Use the currently accepted Latin names, for e.g. Trametes instead of Coriolus, and Lentinula instead of Lentinus
  • Latin names should be in italics
  • Glycosides do not represent a distinct group of compounds, one can find glycosides in several class of natural products i.e. flavonoids, steroids, anthraquinones etc
  • Organic acids and vitamin C are not characteristic to mushrooms, instead I would mention triterpenes, as the major biomarker-type compounds of many fungal species
  • The flavonoid content of mushrooms is a complicated issue, since their regular presence in mushrooms has not been unequivocally proved
  • References no. 25 and 64 are the same
  • The current manuscript should be considerably improved prior a possible publication in this journal

Author Response

For Reviewer 2

We greatly appreciate the efforts of the Editorial Board and reviewers for giving us valuable comments on our manuscript for publication in IJMS. Here, we have enclosed a revised version of the manuscript in response to the reviewer’s comments. Included below is a point-by-point description of our responses to the reviewer’s comments. The revision made in the manuscript was written in blue. We cordially hope that the reviewers find this revised manuscript acceptable for publication in the special issue ‘Role of NF-κB in Carcinogenesis and Its Therapeutic Regulation’ of IJMS

Sincerely,

Hye-Jin Park

Reviewer 3 Report

In this manuscript, Park reviewed the Current application of mushrooms in cancer treatment and their anti-cancer mechanisms. The work is interesting.

1      Table 1 should list all the trials which are completed and with results. e.g. a simple search on the 3rd one in the Table, NCT00970021, the conclusion was “There were no statistically significant differences in treatment response, overall survival, and time to new treatment” doi: 10.1155/2015/718539. You may search https://ichgcp.net/clinical-trials-registry/; and https://clinicaltrials.gov/ etc.

2. For the ongoing trials with no results - you may put in a new Table or present as supplementary data.

3. For the anti-cancer mechanisms, it would be helpful if a graph/diagram is presented.

Reviewer 4 Report

The work presented to me for review, entitled " Current usages of mushrooms in cancer treatment and their anti-cancer mechanisms’’ is a review work. The author once again tries to present the effect of compounds of mushrooms origin in the treatment of cancer. It is not revealing, the more so as there are more precise works in the literature that contain more information necessary for this type of article ( e.g ,,Anticancer Activities of Mushrooms: A Neglected Source for Drug Discovery, ,,Medicinal mushrooms in adjuvant cancer therapies: an approach to anticancer effects and presumed mechanisms of action, ,,Treasures from the forest: Evaluation of mushroom extracts as anti-cancer agents, ,,The Promising Role of Mushrooms as a Therapeutic Adjuvant of Conventional Cancer Therapies, ,,Immunomodulatory effect of mushrooms and their bioactive compounds in cancer: A comprehensive review). Creating a review work should be supported by an earlier review of the literature and possible modification under which the work can be enriched and diversified. Of course I always appreciate the authors' work, but unfortunately it is difficult to find any news here. First, the Introduction is very poor and does not provide any information that should be introduced specifically into the cycle. The following chapters are also not informative and do not contain any news for the reader. The chapters are short, practically random. One of the most sublime elements in the review articles is the time span of the literature on which the authors worked. There is no desirable information here, i.e. the author could use articles from the last year but also from 20 years. There is no information about it. The idea of ​​the review work is the time frame and an interesting approach to the team. One Table that shows the mechanisms of action on several examples, in this type of articles it is quite weak, the more that the author of the Introduction wrote that there are about 700 species. The conclusions are predictable and also lack any concrete information. Summing up, the author should look more closely at the literature and verify what is already there and what could be added or varied. First of all, for the future, it is necessary to introduce sections on the timeframe and selection of literature. Secondly, we should follow some key according to which we want to cover the topic, so as not to repeat what already exists in the literature. I suggest extending the analysis to include selection for in vivo tests on humans or animals, but within certain time limits. I suggest rearranging the work and maybe focusing on specific types of cancer and making selections in this regard, because the form presented by the author is already described and functions in the literature. Finally, I would like to say once again that I appreciate some contribution to the work, but it does not stand out with anything special and does not bring any new information in this field and in this form.

Round 2

Reviewer 1 Report

In my view, the authors made substantial corrections. However, lines 23-24 required appropriate citation. 

Lines 33-41 are duplicates of preceding sentences. 

Line 44-45, the author mentioned many studies but cited only one paper. In such a case, the author should cite more than 5 recent papers (Last 2-3 years).

I couldn't find where Table 1 referred to in the main text. 

Are all the figures showing mechanisms newly drawn/generated? Otherwise, need appropriate citations. 

Authors have given less emphasis to bioactive compounds. It would be appreciated if bioactive compounds were comprehensively reviewed with their extraction techniques and structure. 

Author Response

I greatly appreciate your effort in giving me valuable comments on this manuscript for publication in IJMS. Here, I have enclosed a revised version of the manuscript in response to your comments. Included below is a point-by-point description of my responses to your comments, which are written in red. I cordially hope that you find this revised manuscript acceptable for publication in IJMS. Thank you.

Reviewer 3 Report

I can recommend to accept this revised version for publication.

Author Response

Thank you for accepting this manuscript for publication. 

Reviewer 4 Report

Unfortunately, I am sorry to admit that the author has improved my suggestions on average, so it is more difficult for me to find answers to the reviews, it is only laconically written that the changes are visible in blue. Besides, there is one of the most important things I wrote about, namely the ,,study design'' section which is indispensable in this type of article- that is, the specific time frame from which the author took the articles, and so on.

Author Response

I greatly appreciate your effort in giving me valuable comments on this manuscript for publication in IJMS. Here, I have enclosed a revised version of the manuscript in response to your comments. Included below is a point-by-point description of my responses to your comments, which are written in red. I apologize that some parts of the revision note in response to your comments were written poorly. I cordially hope that you find this revised manuscript acceptable for publication in IJMS. Thank you.
